# High resolution respirometry of isolated mitochondria from adult *Octopus maya* (Class: Cephalopoda) systemic heart

Ana Karen Meza-Buendia[1⊙], Omar Emiliano Aparicio-Trejo[2⊙], Fernando Díaz[1‡], Claudia Caamal-Monsreal[3,4⊙], José Pedraza-Chaverri[5‡], Carolina Álvarez-Delgado[6‡], Kurt Paschke[7,8⊙], Carlos Rosas[3,4⊙]*

**1** Laboratorio de Ecofisiología de Organismos Acuáticos, Departamento de Biotecnología Marina, Centro de Investigación Científica y de Educación Superior de Ensenada (CICESE), Ensenada, Baja California, México, **2** Departamento de Fisiopatología Cardio-Renal, Instituto Nacional de Cardiología "Ignacio Chávez", Mexico City, Mexico, **3** Unidad Multidisciplinaria de Docencia e Investigación, Facultad de Ciencias, Universidad Nacional Autónoma de México, Sisal, Mexico, **4** Laboratorio de Resilencia Costera LANRESC, CONACYT, Sisal, Mexico, **5** Laboratorio F-315, Departamento de Biología, Facultad de Química, Universidad Nacional Autónoma de México, Ciudad de México, Mexico, **6** Departamento de Innovación Biomédica, Centro de Investigación Científica y de Educación Superior de Ensenada (CICESE), Baja California, Mexico, **7** Instituto de Acuicultura, Universidad Austral de Chile, Puerto Montt, Chile, **8** Centro FONDAP de Investigación de AltasLatitudes (IDEAL), Punta Arenas, Chile

⊙ These authors contributed equally to this work.
‡ FD, JPC and CAD also contributed equally to this work.
* crv@ciencias.unam.mx

**Data Availability Statement:** The data underlying the results presented in the study are available

## Abstract

Mitochondrial respirometry is key to understand how environmental factors model energetic cellular process. In the case of ectotherms, thermal tolerance has been hypothesized to be intimately linked with mitochondria capability to produce enough adenosine triphosphate (ATP) to respond to the energetic demands of animals in high temperatures. In a recent study made in *Octopus maya* was proposed the hypothesis postulating that high temperatures could restrain female reproduction due to the limited capacity of the animals' heart to sustain oxygen flow to the body, affecting in this manner energy production in the rest of the organs, including the ovarium Meza-Buendia AK et al. (2021). Unfortunately, until now, no reports have shown temperature effects and other environmental variables on cephalopod mitochondria activity because of the lack of a method to evaluate mitochondrial respiratory parameters in those species' groups. In this sense and for the first time, this study developed a method to obtain mitochondrial respirometry data of adult *Octopus maya*'s heart. This protocol illustrates a step-by-step procedure to get high yield and functional mitochondria of cephalopod heart and procedure for determining the corresponding respiratory parameters. The procedure described in this paper takes approximately 3 to 4 hours from isolation of intact mitochondria to measurement of mitochondrial oxygen consumption.

## Introduction

Mitochondria are essential organelles that control cell life and death and involved in key biosynthetic and catabolic metabolic reactions (oxidation of fatty acids, carbohydrates, and

from: https://zenodo.org/record/6824311#.
Ys3hwDdBy3A.

**Funding:** Initials of the author: CR Grant number:
61503 Funder: Consejo Nacional de Ciencia y
Tecnología URL of Funder: https://conacyt.mx
Initials of the author: CR Grant number: IN203022
Funder: Dirección General de Asuntos del personal
académico, universidad Nacional Autonoma de
México URL of Funder: https://dgapa.unam.mx/
The funders had and will not have a role in study
design, data collection and analysis, decision to
publish, or preparation of the manuscript.

**Competing interests:** The authors have declared
that no competing interests exist.

proteins, amino acids, and purine biosynthesis) besides synthesizing most of the adenosine tri-phosphate (ATP) through oxidative phosphorylation (OXPHOS) to supply the energy demands of the different aerobic tissues [1]. Moreover, mitochondria are also involved in cellular signaling processes that regulate key factors, such as redox homeostasis and apoptosis [2, 3].

For this reason, recent studies point to mitochondrial dysfunction as key a point in the development and progression of various pathophysiological processes related to cellular aerobic energy metabolism [4–10]. Therefore, evaluation of mitochondrial bioenergetics parameters, specifically respiratory states, is important to understand tissue-specific changes in energy metabolism. Currently, new methods have been used that allow evaluating mitochondrial respiratory states in isolated mitochondria, permeabilized cells in homogenized tissue, in intact cells and in tissue sections [11, 12].

These techniques make it possible to evaluate the basic mechanisms of mitochondrial function, such as oxygen consumption, membrane potential, proton leakage, ATP production, and reactive oxygen species (ROS) production [13–18].

Mitochondrial research has benefited from the availability of organelle preparations isolated from tissues, specifically those procedures to isolate mitochondria based on differential centrifugation [19], which allows separating cell constituents based on their different sedimentation characteristics after mechanical tissue homogenization [2]. Once isolated, the functionality of the mitochondria can be studied by adding different reagents or inhibitors that enhance or inhibit, step by step, the processes involved in mitochondrial respiration [20, 21].

Mitochondrial isolation allows a deeper insight into changes in the electron transport system and OXPHOS because the reagents can be added directly to the mitochondria without interference from other cellular components, allowing further breakdown of mitochondrial processes. However, a disadvantage of this procedure is that mitochondrial morphology can be altered, and cellular influence and context are completely lost [11, 12].

Currently many mitochondrial isolation protocols have mostly focused on mammalian tissues, such as liver, heart, brain, adipose tissue, skeletal muscle, and so on [3, 22, 23]. Likewise, considerable progress has been made in the study of mitochondria isolation from yeast [24], cultured cells like liver, HepG2 [25], cell lines such as HeLa [26] or prostate cancer, LNCaP [27].

Despite the great progress in the development of mitochondrial isolation protocols in various tissues, vertebrates are the main animal model due to the interest in the mitochondrial physiology of various pathologies, even fish have been used as research models in neurodegenerative diseases such as Parkinson's [28, 29]. Many other studies have been conducted in recent years to assess the relationship between mitochondrial energy production and environmental stressors in marine invertebrates to predict and understand how environmental changes affect performance and abundance in marine ecosystems and how mitochondrial adaptation may allow animals to live longer [20, 21, 30–33].

Among invertebrates, cephalopods have a wide diversity and distribution in the marine environment, being key species from an ecological and economical point of view [34–39]. Currently, evidence has been reported suggesting they could be particularly sensitive to warming due to their high metabolic rates and energetic demands [37–43]. For this reason, the study of the form in which temperature changes affects cephalopod physiology is a key aspect in researching thermal biology of this marine group [35, 38, 44–47].

The study of marine invertebrate mitochondria began in the early 1960s with a description of their ultrastructure and function in marine species [48–52]. It is worth mentioning that in many of the published mitochondrial isolation protocols the intention of some authors has been to provide a general framework, which can be modified by the researcher according to their objectives. Thus, the procedures differ in speed of the differential centrifugation steps

and in sugars used as osmolytes in mitochondrial isolation buffer. According to Frezza *et al.* [2] small changes in sedimentation rate (600 *vs* 800 g, 7,000 *vs* 8,000) do not affect quality and performance of mitochondrial preparation. The goal of a mitochondrial isolation is to obtain as pure and functional organelles as possible.

Although, Oellermann *et al.* [53] evaluated mitochondrial dynamics of cuttlefish hearts (*Sepia officinalis*), the protocol used was through permeabilization of cardiac fibers, based on Saks *et al.* [54] and did not use isolated mitochondria. This study shows a protocol of mitochondria isolation from the heart tissue of an adult octopus species: *Octopus maya*. The heart is a key organ in vertebrates and invertebrates because of its role in oxygen flow from the gills to peripheral tissues and can be used as a model to determine the functionality of mitochondria to respond to environmental changes [55]. The sequential addition of substrates and inhibitors allows characterizing different respiratory parameters of different segments of the electron transport system in a single experiment. After the isolation procedure, high-resolution respirometry was applied to determine oxygen consumption rates and respiratory states: 1', 2', 3 and 4'o. State 1' is supported by endogenous substrates; State 2' refers to oxygen consumption in the presence of exogenous substrates only; State 3' refers to oxygen consumption in the presence of exogenous substrates and ADP; State 4'o -oligomycin-induced- refers to oxygen consumption after ATPsynthase inhibition [56].

In addition, to settle isolation quality, respiratory control (RC) is obtained–an index of oxygen consumption coupling with ATP production–and calculated as the relationship between states 3' and 4'o. Therefore, this research study describes a protocol to isolate intact and functional *Octopus maya* systemic heart mitochondria by differential centrifugation and to use these isolated mitochondria for functional and bioenergetic studies, such as high-resolution respirometry.

## Materials and methods

The protocol described in this peer-reviewed article is published on protocols.io dx.doi.org/10.17504/protocols.io.yxmvmn3z6g3p/v2 and included for printing as (S1 File) with this article. Also included in supporting information is a schematic of the mitochondrial isolation used in this article (S1 Fig) and a representation of the method used to determine the rate of oxygen consumption in each respiratory state (S2 Fig).

## Expected results

To test the method, a group of four *O. maya* adult males (average weight of 1 348 ± 263.49 g) were obtained from a wild population using the local method called "Gareteo" [57, 58]. Animals were individually conditioned in 80 L tanks with filtered and aerated seawater at 24°C and oxygen levels higher than 5.5 mg/L. During the conditioning period animals were fed twice a day with a crab-squid paste [59]. The procedures were approved by the ethic commission of the Faculty of Science at UNAM (Universidad Nacional Autónoma de México, Comisión de Ética Académica y Responsabilidad Científica /Bioética/25102021).

Using the protocol described, a respiratory control (RC) of 6 was obtained in the adult *Octopus maya*'s heart which reflects the good coupling of the mitochondria, indicating that they have a high capacity for Proline oxidation (see S1 Table), and ATP turnover, as well as low proton leakage (Fig 1).

Moreover, in respiratory parameter of State 3, (obtained after correcting respiratory state 3' with respiratory state ROX; see S1 File), an expected mean value of 869.51 ± 78.90 pmol $O_2$ $s^{-1}$ $mg^{-1}$ was obtained (Fig 2).

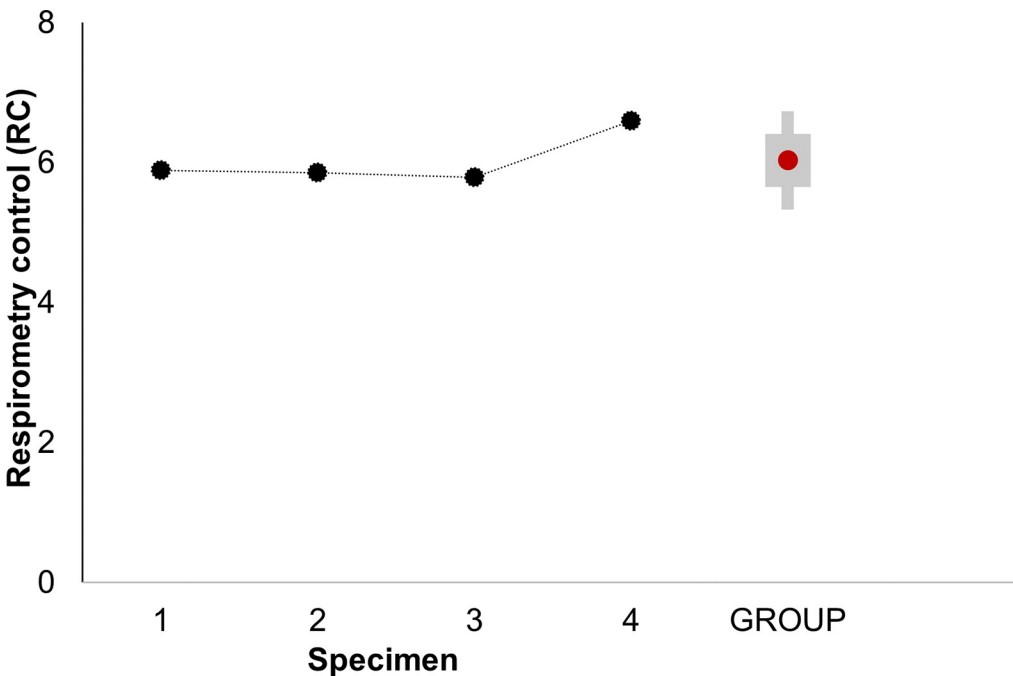

**Fig 1. Respiratory control parameter (RC) of isolated mitochondria from the systemic heart of *Octopus maya*.** The RC of systemic heart mitochondria was obtained from the ratio of respiratory states 3 (S3) and 4o (S4o) once corrected for the respiratory state ROX (S3 = S3'-ROX and S4o = S4'o-ROX). The RC is calculated individually (black dots) for each adult *O. maya* specimens (N = 4, mean weight 1 348 ± 263.49 g) using a mitochondrial concentration of 377± 0.02 μg ml $^{-1}$. GROUP; represents the mean RC value of the four specimens (red dot), ±SD (gray box).

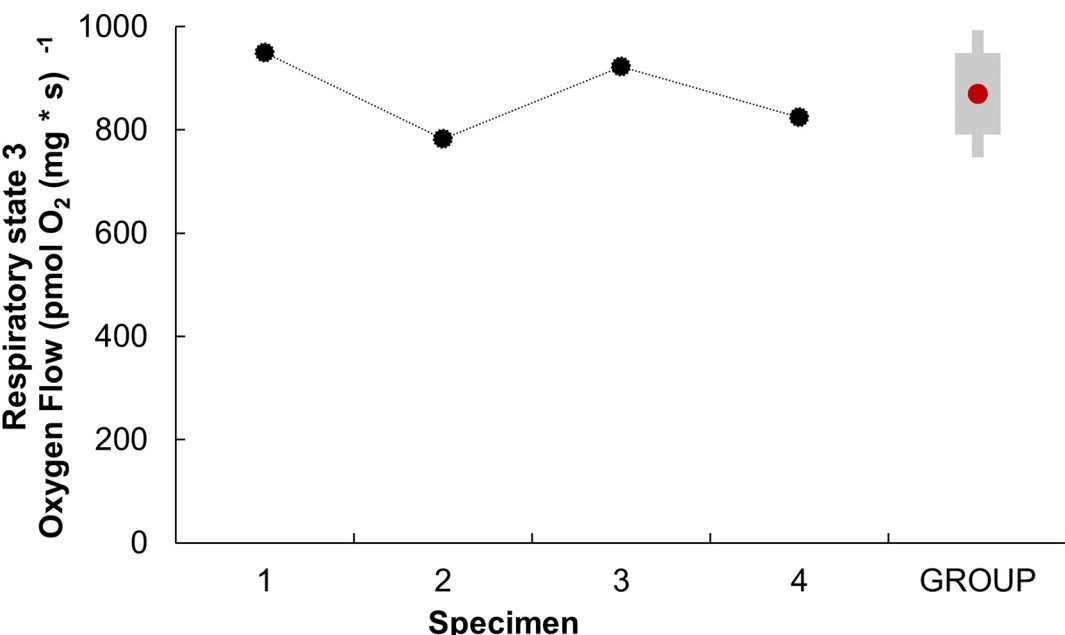

**Fig 2. Respiratory parameter of state 3 (S3) of isolated mitochondria from the systemic heart of *Octopus maya*.** The S3 of the systemic heart mitochondria was obtained from the correction of the respiratory state 3' (S3') and ROX (S3 = S3'-ROX). To obtain S3', adenosine diphosphate (ADP) was added at a final concentration of 1.25 mM. The S3 was obtained individually (black dots) for each adult *O. maya* specimens (N = 4, mean weight 1 348 ± 263.49 g) using a mitochondrial concentration of 377 ± 0.02 μg ml $^{-1}$. GROUP; represents the mean S3 value of the four specimens (red dot), ±SD (gray box).

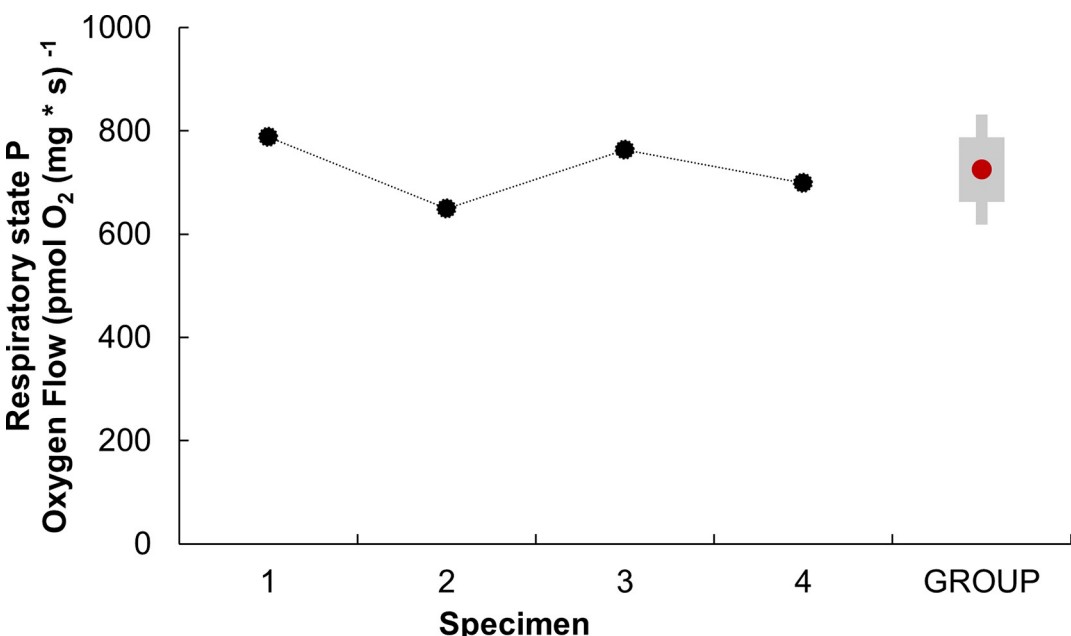

**Fig 3. Phosphorylation state parameter (*P*) of isolated mitochondria from the systemic heart of *Octopus maya*.** *P* of systemic heart mitochondria was obtained by subtracting the respiratory parameter S3 from the respiratory parameter S4o (*P* = S3-S4o; S3 and S4 are ROX-corrected values: S3 = S3'-ROX y S4o = S4'o-ROX, respectively). The P was obtained individually (black dots) for each adult *O. maya* specimens (mean weight 1 348 ± 263.49 g) using a mitochondrial concentration of 377 ± 0.02 μg ml -1. GROUP; represents the mean P value of the four specimens (red dot), +SD (gray box).

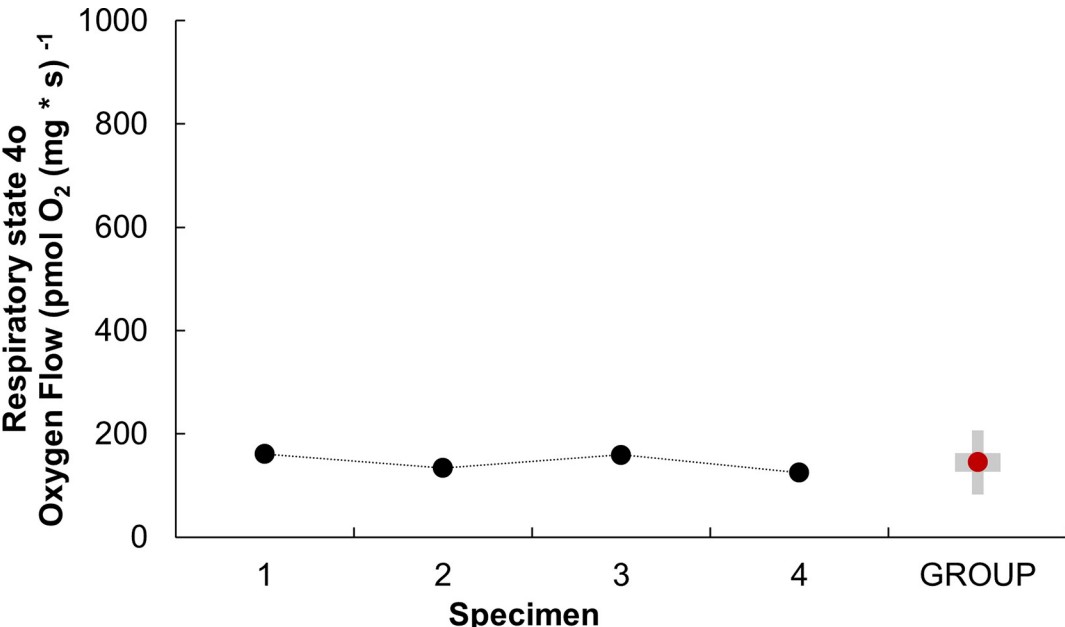

**Fig 4. Respiratory parameter of state 4 (S4o) of isolated mitochondria from the systemic heart of *Octopus maya*.** S4o in *O. maya* systemic heart mitochondria were obtained from the correction of respiratory state 4' oligomycin-induced (S4'o) y ROX: S4o = S4'o-ROX. The S4o was obtained individually (black dots) for each adult *O. maya* specimens (mean weight 1 348 ± 263.49 g) using a mitochondrial concentration of 377 ± 0.02 μg ml $^{-1}$. GROUP; represents the mean S4o value of the four specimens (red dot), +SD (gray box).

In the phosphorylation state parameter (*P*), the mean value was 724.76 ± 62.79 pmol $O_2$ $s^{-1}$ $mg^{-1}$. This oxygen consumption is directly attributed to ATP production (Fig 3; see S1 File for obtaining the P state parameter).

When oligomycin was added (Respiratory parameter of State 4o), a mean value of 144.75 ± 18.17 pmol $O_2$ $s^{-1}$ $mg^{-1}$ was obtained (Fig 4). The above demonstrates that confident and stable data of the metabolism of octopus heart mitochondria can be obtained with this method.

## Supporting information

**S1 File. Step-by-step protocol, also available on protocols.io.**
(PDF)

**S1 Fig. Mitochondrial isolation of a systemic heart from an adult *Octopus maya*.**
(TIF)

**S2 Fig. Schematic representations of the method used to determine the rate of oxygen consumption in each respiratory state (S2', S3', S4'o and ROX).**
(TIF)

**S1 Table. Action and concentration of agents used for measuring mitochondrial respiration of isolated mitochondria from the systemic heart of *Octopus maya*.**
(TIF)

**S2 Table. Rate of oxygen consumption in each respiratory state.**
(TIF)

## Acknowledgments

This work was carried out due to the collaboration between the Centro de Investigación Científica y de Educación Superior de Ensenada, Baja California, and the Universidad Nacional Autónoma de México (UNAM) based in Sisal, Yucatán who provided the facilities, personal and materials necessary to maintain the animals and carry out the experiments. We are grateful to fisherman Antonio (Moluscos del Mayab cooperative), who helped collect the animals used. Thanks to D. Fischer who provided the English edition.

## Author Contributions

**Conceptualization:** Ana Karen Meza-Buendia, Omar Emiliano Aparicio-Trejo, Fernando Díaz, José Pedraza-Chaverri, Carolina Álvarez-Delgado, Kurt Paschke, Carlos Rosas.

**Data curation:** Ana Karen Meza-Buendia, Carlos Rosas.

**Formal analysis:** Ana Karen Meza-Buendia, Omar Emiliano Aparicio-Trejo, Claudia Caamal-Monsreal, Carlos Rosas.

**Funding acquisition:** Carlos Rosas.

**Investigation:** Claudia Caamal-Monsreal, Kurt Paschke.

**Methodology:** Ana Karen Meza-Buendia, Claudia Caamal-Monsreal, Kurt Paschke, Carlos Rosas.

**Supervision:** Omar Emiliano Aparicio-Trejo, Fernando Díaz, José Pedraza-Chaverri, Carolina Álvarez-Delgado, Carlos Rosas.

**Validation:** Omar Emiliano Aparicio-Trejo, Carolina Álvarez-Delgado, Kurt Paschke.

**Visualization:** Fernando Díaz, José Pedraza-Chaverri, Carolina Álvarez-Delgado.

**Writing – original draft:** Ana Karen Meza-Buendia, Carlos Rosas.

**Writing – review & editing:** Carlos Rosas.

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
