## [Decision Letter · Decision Letter 0]

30 May 2022

PONE-D-22-06578High resolution respirometry of isolated mitochondria from adult Octopus maya (Class: Cephalopoda) systemic heartPLOS ONE

Dear Dr. Vázquez,

Thank you for submitting your manuscript to PLOS ONE. After careful consideration, we feel that it has merit but does not fully meet PLOS ONE’s publication criteria as it currently stands. Therefore, we invite you to submit a revised version of the manuscript that addresses the points raised during the review process. Both reviewers have expressed positive opinion about the manuscript. However, they have also requested several changes (including adding more details to the protocol and information to the text, rearrangement of figures and English language editing) that will need to be addressed prior to acceptance of the manuscript.

We look forward to receiving your revised manuscript.

Kind regards,

Metodi D Metodiev, Ph.D.

Academic Editor

PLOS ONE

Journal Requirements:

2. We note you have not provided a Protocol.io PDF version of your protocol. As noted in our submission requirements, please upload a Protocol.io PDF version of your protocol as a Supporting Information file and name the file ‘S1 file’. Please update your Supporting Information Captions if necessary. If you have not yet uploaded your protocol to Protocols.io you are welcome to use the Protocols.io customer service code ‘PLOS2021.’ When using this customer code while submitting to Protocols.io, please make reference to your PLOS ONE submission, including your PLOS ONE manuscript number. With this customer code, Protocols.io editorial staff will import and format your protocol at no charge. For more information, please see our submission guidelines:  https://journals.plos.org/plosone/s/submission-guidelines#loc-guidelines-for-specific-study-types

“This study was partially financed by the Universidad Nacional Autónoma de México (UNAM) throught its Programa de Apoyo a Proyectos de Investigación e Innovación Tecnológica [CR IN 204019] and Consejo Nacional de Ciencia y Tecnología (CONACYT) FORDECYT-PRONACES/61503/2020 grant to CR.”

“Initials of the author: CR

Grant number: 61503

Funder: Consejo Nacional de Ciencia y Tecnología

URL of Funder:

https://conacyt.mx

Initials of the author: CR

Grant number: 204019

Funder: Dirección General de Asuntos del personal académico, universidad Nacional Autonoma de México

URL of Funder: https://dgapa.unam.mx/

The funders had and will not have a role in study design, data collection and analysis, decision to publish, or preparation of the manuscript.”

Reviewers' comments:

Reviewer's Responses to Questions

**Comments to the Author**

1. Does the manuscript report a protocol which is of utility to the research community and adds value to the published literature?

Reviewer #1: Yes

Reviewer #2: Yes

2. Has the protocol been described in sufficient detail?

Descriptions of methods and reagents contained in the step-by-step protocol should be reported in sufficient detail for another researcher to reproduce all experiments and analyses. The protocol should describe the appropriate controls, sample sizes and replication needed to ensure that the data are robust and reproducible.

Reviewer #1: Yes

Reviewer #2: Partly

3. Does the protocol describe a validated method?

Reviewer #1: Yes

Reviewer #2: Yes

4. If the manuscript contains new data, have the authors made this data fully available?

Reviewer #1: Yes

Reviewer #2: Yes

**5. Is the article presented in an intelligible fashion and written in standard English?**

Reviewer #1: Yes

Reviewer #2: Yes

6. Review Comments to the Author

Reviewer #1: Carlos Rosas Vazquez present very thorough protocol to obtain high yield and functional mitochondria of cephalopod heart for the first time and detailed procedure for determining the respiratory parameters. A key feature that is repeated throughout is the success for reproducibility of the protocol for extracting functional mitochondria.

Authors carefully compared available methods for isolation of functional mitochondria that mainly focus on mammalian tissues, yeast, and cell cultures and certainly aware of drawbacks of each method. They also pointed out that despite marine invertebrate mitochondria studies started in early 1960s, studies mostly focused on ultrastructure of mitochondria then mitochondrial dynamics of cuttlefish hearts was an attempt on permeabilized cardiac fibers but not on isolated mitochondria.

The main strength of this paper is that it provides very first data from bioenergetics information on isolated heart mitochondria of a marine invertebrate. As such this article represents a novel study which will certainly contribute to our knowledge about mitochondrial functions and environmental adaptation of the species.

In eukaryotes, the primary hub of energy metabolism is the mitochondria, which conserve the free energy released by the biological oxidation of food-derived substrates (mostly fats and carbohydrates) in the form of a proton gradient across the inner mitochondrial membrane. In turn, this drives the synthesis of ATP, as well as other energy-requiring processes such as metabolite and ion transport, and production of heat. Some organisms (homeotherms) make use of this heat, to maintain their body temperature at an approximately constant level, whereas others (poikilotherms) tolerate a wide range of internal and intracellular temperatures. Some of the weaknesses claimed in the reasoning part of the manuscript are based on the fact that mitochondria sole function is indicated as to produce ATP. Considering the thermal tolerance of the ectotherms tightly linked to mitochondrial capability to adjust the metabolic capacity, authors should also mention the ability of mitochondria in heat production. Another possible criticism would be that based on the hypothesis postulating high temperatures could restrain female reproduction due to limited capacity of the animals’ heart to sustain oxygen flow to the body affecting energy production for the rest of the organs specifically ovaries, authors did not isolate and bioenergetically compare mitochondria from female heart.

Authors provide as main figures Respiratory Control, Respiratory State 3, Phosphorylation State, and Respiratory state 4 data but not individual “rate of oxygen consumption in each respiratory states” graphs where readers could immediately experience the functionality proof of isolated mitochondria. Instead as a supplementary information a schematic representation graph is presented. The sole purpose of the methodological paper is to provide an efficient and functional mitochondria isolated from the invertebrate heart therefore providing direct proof in the main figures would be more useful.

In the last sentence of the abstract section authors linked the amount of time required for isolation and confident and reproducibility of the result which should not be corelated, i.e, reproducible and confident results should be independent the amount of time required to isolate functional mitochondria.

The authors should revise the language to improve readability.

Reviewer #2: The manuscript provides a manual for mitochondria isolation and high-resolution respirometry analyses on Octopus maya systemic heart. This may be of interest for researchers working with the model and in general for researchers working with mitochondrial physiology, as many of the instructions detailed in the manuscript may be of use for analyses of any organism.

Because other detailed high-resolution respirometry protocols are available in the literature, I would suggest that the authors further stress the particularities, if any, when working with their model. I think the manuscript would benefit from some experimental results. Have the authors got the chance for instance to compare mitochondria performance from organisms living in different conditions?

In general, I find the text well written, although I would recommend a thorough revision of English usage throughout the manuscript because there are some typos to be corrected and expressions that may need clarification.

Specific comments:

MAIN TEXT:

- Line 65 ‘as a key point’

- Line 81 ‘all the processes that occur in this organelle’ I think the authors do not really mean all the processes can be monitored. Please rephrase

- Line 95-97 Please review English usage

- Line 118 Perhaps the authors meant ‘The goal of a mitochondrial isolation is to obtain as pure and functional organelles as possible’.

- Line 128 Perhaps the authors meant ‘of different segments’ instead of ‘or different segments’

- Line 140 Perhaps the authors meant ‘and to use’ instead of ‘and use’

- Line 155: Can the authors provide any reference of what would be a ‘good’ respiratory ratio?

- Figure 1 and 2: Are the results mean of how many independent recordings? Can the authors provide any variation /standard deviation data for each point?

The authors mention the experiments are results from six independent animals but the graphs show results for 4 individual specimens. Please clarify.

- Figure 3: legend for x-axis is missing

- Line 215 Please correct the typo: ‘throught’

SUPPLEMENTARY INFO:

- All centrifugation steps should be given in rcf or g

- When mM concentration is too small please change the units to micromolar (i. e. oligomycin, antimycin, …)

- Table1/2: Since BSA can not be expressed in mM. Could it be added in a separate row below osmole values with its correct units?

Also, can they specify when the BSA is to be added?

- Step 5 of the Preparation of MiR05 mitochondrial respiratory buffer: Do the authors mean it takes up for 90 minutes to stabilise the pH after KOH addition? Please clarify this point

- Preparation of the 500 mM ADP stock solution: How do you check pH on such a small volume? Do you use a method other than a usual electrode?

- Step 7 of the Isolation of mitochondria from the systemic heart of adult octopus: perhaps it should state ‘keep on ice for 5 min’ instead of ‘kept on ice for 5 min’?

- Step 9 of the Isolation of mitochondria from the systemic heart of adult octopus: I think there is no need to introduce the term ‘pelletizing’ that is not commonly seen in the literature and it is not used anywhere else in the manuscript

- Step 10 of the Isolation of mitochondria from the systemic heart of adult octopus: What the authors mean by ‘no dyes’ here?

- Step 15 of the Isolation of mitochondria from the systemic heart of adult octopus: Can the authors provide with any reference or range of the expected typical yield for the mitochondrial prep?

- Step 4: Substrate/Inhibitor titration analysis. The authors describe a very specific amount of mitochondria to be used. Can they provide with a range instead, or make it clear that 565 μg ml -1 is the amount used for the example experiment? I am guessing the recommended range is stated at the end of the paragraph…

- Step 5: Substrate/Inhibitor titration analysis. ‘The corresponding respiratory substrates must be immediately added to avoid mitochondrial membrane potential depolarization’. Can the authors explain what do they refer to more clearly? What are the options for respiratory substrates to be used?

- Step 6: Substrate/Inhibitor titration analysis. Can the authors provide a range of expected increase based on their observations?

- Step 8: Substrate/Inhibitor titration analysis. ‘The rate of oxygen consumption should be faster than the rate of consumption observed when adding the substrate alone, indicating that well-coupled mitochondria have been obtained’. Can the authors provide a range of the expected respiratory control/phosphorylation state ratios?

- Step 10: Substrate/Inhibitor titration analysis: ‘and the rate of oxygen consumption begins to drop rapidly until the rate of consumption is constant’. Do the authors mean ‘and the rate of oxygen consumption drops rapidly until reaching a plateau’?

- Step 10: Substrate/Inhibitor titration analysis: I suggest the authors use the present tense if these are general recommendations.

- Supplementary Figure 1 : Can the green lines showing the addition of the different reagents be lengthened so that all the green signs can be correctly visualized? Can they also mark states 1 and 2 in the graph?

- Supplementary figure 2 Mitochondrial isolation of a systemic heart from an adult Octopus maya: This figure is incorrectly labeled as supplementary figure 1 in the text

- Supplementary Table 1: Antimycin and oligomycin could be described as inhibitors of complex III and ATP synthase to match phrasing used for Rotenone description

- Supplementary Figure 2 is not listed in the main manuscript text (line 207 and the following). None of the supplementary figures is referenced in the main text.

7. PLOS authors have the option to publish the peer review history of their article (what does this mean?). If published, this will include your full peer review and any attached files.

Reviewer #1: **Yes: **Mugen Terzioglu

Reviewer #2: No

---

## [Author Response · Author response to Decision Letter 0]

13 Jul 2022

Response to Reviewers

Reviewer #1:

In the context of the comment from:

1. In eukaryotes, the primary hub of energy metabolism is the mitochondria, which conserve the free energy released by the biological oxidation of food-derived substrates (mostly fats and carbohydrates) in the form of a proton gradient across the inner mitochondrial membrane. In turn, this drives the synthesis of ATP, as well as other energy-requiring processes such as metabolite and ion transport, and production of heat. Some organisms (homeotherms) make use of this heat, to maintain their body temperature at an approximately constant level, whereas others (poikilotherms) tolerate a wide range of internal and intracellular temperatures. Some of the weaknesses claimed in the reasoning part of the manuscript are based on the fact that mitochondria sole function is indicated as to produce ATP’. 

Thank you for the comment. The objective of this paper was the implementation of a method to isolate intact mitochondria of octopus species (currently with few published papers), which are used to assess their functionality. This manuscript makes this method available so that it can be implemented in other research focused on understanding how environmental factors, in this case, temperature, would modulate energy generation processes through the ATP production (90% of cellular ATP comes from mitochondria in animals). Although we do not directly measure ATP production from mitochondria, we obtained an indirect measure of this process, through respiratory parameter P obtained with this method. Besides the production of ATP, we know that mitochondria not only perform this critical function since they are known as the hallmark of eukaryotic life and are involved in all essential cellular processes (Lane 2005). Please see lines 62 to 68.

References

Lane N. Power, sex, suicide: mitochondria and the meaning of life. New York (NY): Oxford University Press. 2005. p. 368.

Sokolova I. Mitochondrial Adaptations to Variable Environments and Their Role in Animals’ Stress Tolerance. Integr Comp Biol. 2018; 58(3): 519–531. https://doi.org/10.1093/icb/icy017

2. Considering the thermal tolerance of the ectotherms tightly linked to mitochondrial capability to adjust the metabolic capacity, authors should also mention the ability of mitochondria in heat production.

Thank you for the comment. The reason we do not mention the heat-producing capacity of mitochondria in ectotherms is because their function is not necessarily this. In general, as mitochondria oxidize substrates to produce ATP, they tend in parallel to release heat as a by-product of catabolic reactions, so the utilization of this heat will differ between whether an organism is ectothermic or not (endothermic). Endotherms have the metabolic capacity to accelerate heat production mechanisms that they require because they can raise their body temperature above ambient temperature. However, heat generation in ectotherms diffuses too fast to be used as a natural heat source.

Reference

Macherel D, Haraux F, Guillou H, Bourgeois O. The conundrum of hot mitochondria. Biochim Biophys Acta Bioenerg. 2021; 1862(2):148348. https://pubmed.ncbi.nlm.nih.gov/33248118/

3. Another possible criticism would be that based on the hypothesis postulating high temperatures could restrain female reproduction due to limited capacity of the animals’ heart to sustain oxygen flow to the body affecting energy production for the rest of the organs specifically ovaries, authors did not isolate and bioenergetically compare mitochondria from female heart.

Thank you for the comment. As you can see, we added the citation of the study mentioned in the abstract as " Recent study..." Citing in the abstract is not a common practice, but maybe it could be more informative to readers that there is a background that supports the necessity of a method to test several hypotheses where energy and mitochondria are involved.

4. Authors provide as main figures Respiratory Control, Respiratory State 3, Phosphorylation State, and Respiratory state 4 data but not individual “rate of oxygen consumption in each respiratory states’ graphs where readers could immediately experience the functionality proof of isolated mitochondria. Instead as a supplementary information a schematic representation graph is presented. ‘The sole purpose of the methodological paper is to provide an efficient and functional mitochondria isolated from the invertebrate heart therefore providing direct proof in the main figures would be more useful.’

Thank you for the comment. We have added a table with the oxygen consumption values for each respiratory state, which we attach to the supporting information of the paper. Please see: Supplementary 2 Table. Oxygen consumption rate in each respiratory state; S2’= State 2’; S3’ = State 3’; S4’o = state 4 oligomycin-induced; ROX= residual non- mitochondrial respiration.

5. In the last sentence of the abstract section authors linked the amount of time required for isolation and confident and reproducibility of the result which should not be corelated, i.e. reproducible and confident results should be independent the amount of time required to isolated functional mitochondria.

Thank you for the comment. we modified the last paragraph of the abstract section. In this modification we mention that the complete procedure described in this work requires 3 to 4 hours. In the previous version where we mentioned that the isolation procedure requires 2 hours, referring only to time to obtain intact systemic cardiac mitochondria. Please see line 55-57, Revised Manuscript with Track Changes.

6. ‘The authors should revise the language to improve readability.’

Thank you for the comment. The language of the manuscript was revised by a specialized editor.

Reviewer #2:

1. Because other detailed high-resolution respirometry protocols are available in the literature, I would suggest that authors further stress the particularities, if any, when working with their model.

Thank you for your comment. Regarding the particularities of the method presented here, we could refer to the use of a mitochondrial isolation buffer with the appropriate osmolarity (826 mOsmoles). The osmolarity of Octopus vulgaris and O. maya has been reported to be ~1170 mOsm/kg H2O and 1150 + 27, respectively (Wells and Wells 1989; Pascual et al, 2019), while Sepia ofﬁcinalis and Loligo forbesi report osmolarities of 1160 and 1032 mOsm/kg H2O, respectively (Kirschner 1991). In consequence, the buffer used now allow obtain functional mitochondria for respirometry measurements. It is important to mention that Oellermann et al. (2012) worked with cardiac fibers from cuttlefish heart, using a different method which consisted of permeabilizing the cardiac fibers without isolating mitochondria. Another distinctive feature of the method is the establishment of the concentrations of the appropriate substrates and inhibitors that allowed us to evaluate the functionality of Octopus maya cardiac mitochondria, which will have potential for future research to evaluate their functionality under different stressors.

Wells MJ, Wells J. Water uptake in a Cephalopod and the function of the so-called ‘pancreas’. J Exp Biol. 1989. 145:215–226.

Kirschner LB. Water and ions. In: Prosser CL, editor. Comparative animal physiology,environmental and metabolic animal physiology. New York (NY): Wiley-Liss; 1991. p. 13–107.

Oellermann M, Pörtner HO, Mark FC. Mitochondrial dynamics underlying thermal plasticity of cuttlefish (Sepia officinalis) hearts. J Exp Biol. 2012; 215(17):2992-3000. https://doi.org/10.1242/jeb.068163

Pascual C, Mascaró M, Rodríguez-Canul R, Gallardo P, Sánchez-Arteaga A, Rosas C, Cruz-López H (2019) Sea surface temperature modulates physiological and immunological condition of Octopus maya. Frontiers in Physiology 10: 739, Doi: 710.3389/fphys.2019.00

2. I think the manuscript would benefit from some experimental results. Have the authors got the chance for instance to compare mitochondria performance from organisms living in different conditions?

Thank you for the comment. Yes, we are making another studies where mitochondrial metabolism has been evaluated. That data will be part of another paper where the experimental design and other results will be shown. As a background for future papers, the present paper is dedicated only to the method, as is required in this type of papers in PlosOne Journal.

3. In general, I find the text well written, although I would recommend a thorough revision of English usage throughout the manuscript because there are some typos to be corrected and expressions that may need clarification.

Thank you for the comment. Typographical errors have been corrected and the use of English has been revised.

Specific comments:

MAIN TEXT

Line 65. Changed 'as key point' to 'as a key point' (Line 69; Revised Manuscript with Track Changes). Thanks for your comment.

Line 81. ‘all the processes that occur in this organelle’ I think the authors do not really mean all the processes can be monitored. Please rephrase.

We appreciate the comment and clarify as follows.

In the context of the text, it said: 

Line 80-82: ‘Once isolated, mitochondria can be studied by adding different reagents or inhibitors that enhance or inhibit, step by step, all the processes that occur in this organelle [20,21]’. This was changed to: 

‘Once isolated, the functionality of the mitochondria can be studied by adding different reagents or inhibitors that enhance or inhibit, step by step, the processes involved in mitochondrial respiration’. Please see line 84-86, Revised Manuscript with Track Changes.

Line 95-97 Please review English usage

Thank you very much for your observation. For better understanding it was changed to (Please see line 98-101, Revised Manuscript with Track Changes): ‘Despite the great progress in the development of mitochondrial isolation protocols in various tissues, vertebrates are the main animal model due to the interest in the mitochondrial physiology of various pathologies, even fish have been used as research models in neurodegenerative diseases such as Parkinson's.’

Line 118 Perhaps the authors meant ‘The goal of a mitochondrial isolation is to obtain as pure and functional organelles as possible’ 

Thank you for your observation. The reviewer's suggestion (Please see line 123, Revised Manuscript with Track Changes) was accepted.

Line 128 Perhaps the authors meant ‘of different segments’ instead of ‘or different segments’ 

Thank you for your observation. The reviewer's suggestion (Please see line 132, Revised Manuscript with Track Changes) was accepted.

Line 140 Perhaps the authors meant ‘and to use’ instead of ‘and use’

Thank you for your observation. The reviewer's suggestion (Please see line 144, Revised Manuscript with Track Changes) was accepted.

Line 155 Can authors provide any reference of what would be a ‘good’ respiratory ratio? 

By definition, the respiratory control ratio is the ratio of mitochondrial respiration that supports ATP synthesis to that needed to compensate for proton leakage. This index or proportion is useful to determine the integrity of mitochondrial preparations, because usually when mitochondria are isolated from cells, some break and these are uncoupled (electron transport occurs in the absence of production of ATP). A high RCR indicates good function and a low RCR usually indicates dysfunction. The utility of RCR is based on its complexity, any change in oxidative phosphorylation will change the RCR, so there is no absolute value of RCR, because they are substrate and tissue dependent. Although the RCR value depends on numerous factors related to oxidative phosphorylation, determining an RCR value for the cardiac mitochondria of Octopus maya, according to the literature, is a bit complex. This is due to that in different animals and organs different substrates and methods are used to evaluate the RCR, making difficult to define a “good” RCR value. However, our method established that an RCR of around six reflects the efficient oxidizing of proline, with a high turnover of ATP and low proton leakage. 

Brand MD, Nicholls DG. Assessing mitochondrial dysfunction in cells. Biochem J. 2011 Apr 15;435(2):297-312. doi: 10.1042/BJ20110162. Erratum in: Biochem J. 2011 Aug 1;437(3):575. PMID: 21726199; PMCID: PMC3076726.

Salin K, Villasevil EM, Anderson GJ, Selman C, Chinopoulos C, Metcalfe NB. The RCR and ATP/O Indices Can Give Contradictory Messages about Mitochondrial Efficiency. Integr Comp Biol. 2018 Sep 1;58(3):486-494. doi: 10.1093/icb/icy085. PMID: 29982616.

Figure 1 and 2 Are the results mean of how many independent recording? Can the authors provide any variation /standard deviation data for each point? The authors mention the experiments are results from six independent animals, but the graphs show results for 4 individual specimens. Please clarify.

Thank you for the comment. The mean of each of the respiratory parameters indicated in the results section was calculated with 4 animals. Mitochondria were isolated from the systemic heart (whole organ was used) of each animal. The mitochondria from each animal were used to evaluate their oxygen consumption in the high-resolution respirometer to test each of the respiratory states. Therefore, the graphs do not show any measure of dispersion, because each animal is represented independently on the x-axis, with a total of 4 individuals. Therefore, to clarify this, graphs 1, 2, 3 and 4 were modified (Please see section Expected results in Revised Manuscript with Track Changes) including mean + SD values for all measurements.

Figure 3: Legend for x-axis is missing 

Thank you for your observation. The missing x-axis legend was added

Line 215 Please correct the typo: ‘throught’ 

Thank you for your observation. Corrected typo. However, the Acknowledgments section (Please see line 217-224, Revised Manuscript with Track Changes) was changed.

SUPPLEMENTARY INFO:

• All centrifugation steps should be given in rfc or g

Thank you for your observation. The change of units from rpm to rfc was made.

• When mM concentration is too small please change the units to micromolar (i.e. oligomycin, antimycin…)

Thank you for your observation. The change of units to micromolar was made in the concentrations that were small.

Rotenone 2.5 micromolar

Antimycin 12.5 micromolar

Oligomycin 2.5 micromolar.

• Table 1/2: Since BSA cannot be expressed in mM. Could it be added in a separate row below osmole values with its correct units? Also, can they specify when the BSA is to be added?

A change has been made, please see support information file table 1 and lines 31-53.

• Step 5 of the Preparation of MiR05 mitochondrial respiratory buffer: Do the authors mean it takes up for 90 minutes to stabilize the pH after KOH addition? Please clarify this point.

Thank you for the comment. Yes, the preparation of this mitochondrial respiration buffer was carried out according to the suggestions of the manufacturer (Gnaigner et al. 2018), who provides the instructions as well as the composition and the reagents to be used.

• Preparation of the 500 mM ADP stock solution: How do you check pH on such as small volume? Do you use a method other than a usual electrode? 

I appreciate your question, to clarify this point. Yes, to check the pH, a Jenco 6230N brand electrode was used.

• Step 7 of the Isolation of mitochondria from the systemic heart of adult octopus: perhaps it should state ‘keep on ice for 5 min’ instead of ‘kept on ice for 5 min’?

Thank you for your observation. Reworded this step because the instruction was not being clear. As follows: ‘Transfer the homogenate by decantation to a pre-cooled 2-ml Eppendorf tube® and centrifuge at 392 rfc at 4 °C for 5 minutes. If the centrifuge is not nearby, keep the tube with the homogenate cold’. Please see support information file, line 119.

• Step 9 of the Isolation of mitochondria from the systemic heart of adult octopus: I think there is no need to introduce the term ‘pelletizing’ that is not commonly seen in the literature and it is no used anywhere else in the manuscript.

Thank you for your comment. Although it is not common to find it in the literature, we suggest keeping the original word “pellet” because we will refer to obtaining an aggregation of mitochondria that we call 'mitochondrial pellet formation'. Please see support information file, line 127.

• Step 10 of the Isolation of mitochondria from the systemic heart of adult octopus: What the authors mean by ‘no dyes’ here?

In this case when mentioning 'non-dyes', we will refer to the fact that there are brushes with different types of bristles, in some type of dye is applied and to avoid any interference of any chemical with the sample it is better to work with brushes with natural fiber bristles or dye-free. Please see support information file, line 130.

• Step 15 of the Isolation of mitochondria from the systemic heart of adult octopus: Can the authors provide with any reference or range of the expected typical yield for the mitochondrial prep?

Thanks for your observation. Some mitochondria isolation methods are found in the literature where typical yields are mentioned, mainly with commercial kits, (i.e. the Thermo Scientific™ Mitochondria Isolation Kit for cultured cells; catalog number 89874). With this kit, a typical yield of approximately 50 µg of protein from 20 million cells after mitochondrial isolation and lysis can be obtained. In our work, the average yield for isolating octopus systemic heart mitochondria is approximately 14 mg of protein/ml per gram of minced tissue (Please see systemic heart protein in the mitochondrial isolation database).

• Step 4: Substrate/Inhibitor titration analysis. The authors describe a very specific amount of mitochondria to be used. Can they provide with a range instead, or make it clear µg ml-1 is the amount used for the example experiment?

Thank you for your observation, which helps us to clarify the use of the amount of mitochondrial protein. In our work, a 2 ml respirometric chamber is used, where a concentration of total mitochondrial protein between in the range of 701-810 µg is injected. With this method it is possible to obtain a final concentration of 350 to 405 µg/ml. According to our results this mitochondria concentration is adequate to carry out respirometry experiments for octopus hearts. 

• Step 5: Substrate/Inhibitor titration analysis. ‘The corresponding respiratory substrates must be immediately added to avoid mitochondrial membrane potential depolarization’ Can the authors explain what do they refer to more clearly?

When we mention the depolarization of the mitochondrial membrane, we refer to the loss of viability of the mitochondria. Although at the moment of isolating mitochondria from cardiac cells (from Octopus maya), they contain endogenous substrates, these tend to be depleted upon contact with oxygen once inside the respirometric chamber. Therefore, the appropriate substrates (in this case proline) should be added immediately after adding the mitochondria, to avoid depolarization of the mitochondrial membrane potential and loss of mitochondrial viability. It is also worth mentioning that at temperatures close to 4°C, depolarization of the mitochondrial membrane is prevented by slowing enzyme activity and leakage, hence the importance of keeping the mitochondrial isolation cold and performing the experiments during the first 4 hours.

Zorova LD, Popkov VA, Plotnikov EY, Silachev DN, Pevzner IB, Jankauskas SS, Babenko VA, Zorov SD, Balakireva AV, Juhaszova M, Sollott SJ, Zorov DB. Mitochondrial membrane potential. Anal Biochem. 2018; 552:50-59. 10.1016/j.ab.2017.07.009.

What are the options for respiratory substrates to be used?

Oellermann et al. (2012) reported the use of succinate and pyruvate, and proline for a cuttlefish species (Sepia officinalis). In the case of octopus species more research will be done to explore the use of other than proline used in the present method.

Oellermann M, Pörtner HO, Mark FC. Mitochondrial dynamics underlying thermal plasticity of cuttlefish (Sepia officinalis) hearts. J Exp Biol. 2012; 215(17):2992-3000. https://doi.org/10.1242/jeb.068163

• Step 6: Substrate/Inhibitor titration analysis. Can the authors provide a range of expected increase based on their observations?

Thank you for the comment. We added a paragraph indicating the minimum value to be expected in this step

Lines 207 – 209: Observe if the oxygen consumption reaches a value of 10 pmol/s/mg Protein. This minimum value was considered an indicator of functional mitochondria 

• Step 8: Substrate/Inhibitor titration analysis. ‘The rate of oxygen consumption should be faster than the rate of consumption observed when adding the substrate alone, indicating that well-coupled mitochondria have been obtained’. Can the authors provide a range of the expected respiratory control/phosphorylation state ratios?

Thank you for the comment. A paragraph was included in this step. 

Lines 216 -221: For respiratory state 2' which is when mitochondria and substrate only are added, one would expect an oxygen consumption rate of 212.51 + 23.09 pmol O₂ s⁻¹ mg⁻¹). Whereas oxygen consumption in the respiratory state 3' (addition of ADP at a final concentration of 1.25 mM) tends to rise 3.5 times the respiratory state 2' (953.51 + 81.48 pmol O₂ s⁻¹ mg⁻¹).

• Step 10: Substrate/Inhibitor titration analysis: ‘and the rate of oxygen consumption begins to drop rapidly until the rate of consumption is constant’ Do authors mean ‘and the rate of oxygen consumption drops rapidly until reaching a plateau?

Thank you for your comment. The paragraph was changed according to your suggestion. If this is correct, by saying that the rate of oxygen consumption is constant we mean that it enters a steady state, being observed on the graph by the beginning of the formation of a plateau (red line).

• Step 10: Substrate/Inhibitor titration analysis: I suggest the authors use the present tense if these are general recommendations.

Thank you for your comment. We made the change of tense in the sentence.

• Supplementary Figure 1: Can the green lines showing the addition of the different reagents be lengthened so that all the green signs can be correctly visualized? Can they also mark states 1 and 2 in the graph? 

The graph was modified for a better visualization of the addition of the reagents. In addition, the respiratory state 2' (S2') was added, however the respiratory state 1' (S1') was not added because the transition between S1' and S2' must be immediate in order not to lose the viability of the mitochondria during respirometry. Mitochondria, when isolated from cells, have their own substrates, however these tend to end and begin to lose functionality, thus S1' is not considered physiologically relevant.

• Supplementary figure 2: Mitochondrial isolation of a systemic heart from and adult Octopus maya: This figure is incorrectly labeled as supplementary figure 1 in the text.

Thank you for your observation. It was named correctly both in the supporting information list and when the figure was attached to the supporting information file.

Supplementary figure Table 1: Antimycin and oligomycin could be described as inhibitors of complex III and ATP synthase to match phrasing used for Rotenone description.

Thank you very much for your comment. It has been changed as the reviewer suggested.

• Supplementary Figure 2 is not listed in the main manuscript text (line 207 and the following). None of the supplementary figures is referenced in the main text.

Thank you for your observation. The list of supporting information was reorganized and a table called: ´S2 Table was created. Oxygen consumption rate in each respiratory state.

---

## [Editor Report · Decision Letter 1]

26 Jul 2022

PONE-D-22-06578R1High resolution respirometry of isolated mitochondria from adult Octopus maya (Class: Cephalopoda) systemic heartPLOS ONE

Dear Dr. Vázquez,

Thank you for submitting your manuscript to PLOS ONE. After careful consideration, we feel that it has merit but does not fully meet PLOS ONE’s publication criteria as it currently stands. Therefore, we invite you to submit a revised version of the manuscript that addresses the points raised during the review process.

We look forward to receiving your revised manuscript.

Kind regards,

Metodi D Metodiev, Ph.D.

Academic Editor

PLOS ONE

Journal Requirements:

Additional Editor Comments:

The authors have addressed all comments from the two reviewers, and I can recommend that the manuscript is accepted for publication. However, I would just like to point out that the centrifugal force abbreviation is RCF (Relative Centrifugal Force), not RFC as is stated in the protocol. The authors should correct this before publication.

---

## [Author Response · Author response to Decision Letter 1]

5 Aug 2022

As can be seen, references were revised and only one citation was removed (Bradford) because this citation was included in the protocol (file: S1 step by step protocol). The RFC was changed by RCF.

---

## [Editor Report · Decision Letter 2]

11 Aug 2022

High resolution respirometry of isolated mitochondria from adult Octopus maya (Class: Cephalopoda) systemic heart

PONE-D-22-06578R2

Dear Dr. Vázquez,

We’re pleased to inform you that your manuscript has been judged scientifically suitable for publication and will be formally accepted for publication once it meets all outstanding technical requirements.

Kind regards,

Metodi D Metodiev, Ph.D.

Academic Editor

PLOS ONE
---

## [Editor Report · Acceptance letter]

19 Aug 2022

PONE-D-22-06578R2 

High resolution respirometry of isolated mitochondria from adult Octopus maya (Class: Cephalopoda) systemic heart 

Dear Dr. Rosas:

I'm pleased to inform you that your manuscript has been deemed suitable for publication in PLOS ONE. Congratulations! Your manuscript is now with our production department. 

Kind regards, 

on behalf of

Dr. Metodi D Metodiev 

Academic Editor

PLOS ONE